# Effects of Direct-Acting Antiviral Agents on the Mental Health of Patients with Chronic Hepatitis C: A Prospective Observational Study

**DOI:** 10.3390/brainsci10080483

**Published:** 2020-07-27

**Authors:** Michele Fabrazzo, Rosa Zampino, Martina Vitrone, Gaia Sampogna, Lucia Del Gaudio, Daniela Nunziata, Salvatore Agnese, Anna Santagata, Emanuele Durante-Mangoni, Andrea Fiorillo

**Affiliations:** 1Department of Psychiatry, University of Campania “Luigi Vanvitelli”, Largo Madonna delle Grazie 1, 80138 Naples, Italy; gaia.sampogna@gmail.com (G.S.); lucia.delgaudio.ldg@gmail.com (L.D.G.); daniela.nunziata84@gmail.com (D.N.); agnesesalvatore@gmail.com (S.A.); andrea.fiorillo@unicampania.it (A.F.); 2Division of Internal Medicine, Unit of Infectious and Transplant Medicine, University of Campania “L. Vanvitelli”, AORN Ospedali dei Colli, Monaldi Hospital, Piazzale Ettore Ruggieri, 80131 Naples, Italy; rosa.zampino@unicampania.it (R.Z.); martina.vitrone@yahoo.it (M.V.); hanna1985@libero.it (A.S.); emanuele.durante@unicampania.it (E.D.-M.); 3Internal Medicine, Department of Advanced Medical and Surgical Sciences, University of Campania “Luigi Vanvitelli”, Piazza Miraglia 1, 80138 Naples, Italy

**Keywords:** chronic hepatitis C, direct-acting antiviral agents, hepatitis C virus, consultation-liaison psychiatry, depression, anxiety

## Abstract

In chronic hepatitis C (CHC) patients, interferon-based treatments showed toxicity, limited efficacy, and psychiatric manifestations. Direct-acting antiviral (DAA) agents appeared safer, though it remains unclear if they may exacerbate or foster mood symptoms in drug-naïve CHC patients. We evaluated 62 CHC patients’ mental status, before and 12 weeks after DAA therapy, by assessment scales and psychometric instruments. We subdivided patients into two groups, CHC patients with (Group A) or without (Group B) a current and/or past psychiatric history. After DAA treatment, Group A patients showed low anxiety and improved depression, no variation in self-report distress, but worse general health perceptions. No significant difference emerged from coping strategies. Depression and anxiety improved in Group B, and no change emerged from total self-reported distress, except for somatization. Moreover, Group B increased problem-focused strategies for suppression of competing activities, and decreased strategies of instrumental social support. Contrarily, Group B reduced significantly emotion-focused strategies, such as acceptance and mental disengagement, and improved vitality, physical and social role functioning. DAA therapy is safe and free of hepatological and psychiatric side effects in CHC patients, regardless of current and/or past psychiatric history. In particular, patients without a psychiatric history also remarkably improved their quality of life.

## 1. Introduction

Hepatitis C virus (HCV) infection is a major cause of chronic liver disease worldwide, liable to progress to cirrhosis and hepatocellular carcinoma (HCC) [1]. HCV infection is frequently associated with extrahepatic manifestations, including neurological and psychiatric complications due to direct viral neurotoxicity or indirect mechanisms, such as cerebral or systemic inflammation, changes of metabolic pathways in infected cells, or neurotransmitter circuits [2,3].

Patients with HCV infection report a higher prevalence of psychiatric disorders, including substance abuse (36%) and mood disorders (28%), compared with the general population [4]. In particular, the HCV course is characterized by medical manifestations, such as fatigue and weakness, but also by several neurological and psychiatric symptoms, including cognitive dysfunctions, sleep disorders, depression, anxiety, and anger/hostility, with a negative impact on patients’ quality of life (QoL). An inadequate patient insight into the disease also contributes to a considerable worsening of health-related QoL [5,6,7].

The interferon-based treatments used formerly showed limited efficacy and high toxicity, as well as a remarkable incidence of psychiatric manifestations [8,9,10]. The introduction of direct-acting antiviral (DAA) agents to treat CHC has led to improved recovery rates (>90% in all genotypes), as well as to prevent disease progression and adverse effect profiles [11], even in CHC patients with previous psychiatric and substance abuse disorders [12]. HCV clearance may indeed prevent liver disease progression and partially control extrahepatic manifestations (i.e., mixed cryoglobulinemia), though not abolish the risk of HCC development [13].

Only a few studies have explored the effects of DAA treatment on CHC patients’ QoL and mental complications. Most data emerged from patient-reported outcomes, mainly obtained during observational trials on tolerance and adherence to treatment [12,13,14]. Other studies [15,16,17], instead, have reported psychiatric symptoms after DAA therapy in CHC patients with a psychiatric history. What remains unclear is whether DAA therapy may exacerbate mood symptoms in patients with prior and/or current psychiatric history or foster psychopathological symptoms in DAA-naïve CHC patients. To this aim, we analyzed mental disturbances in a CHC cohort subgrouped into patients with or without current or past psychiatric history before and 12 weeks after DAA therapy. We also assessed whether DAA treatment influenced the severity of psychiatric symptoms, coping strategies, and QoL.

## 2. Materials and Methods

### 2.1. Subjects

We carried out the present prospective observational study in the Departments of Internal Medicine and Psychiatry, University of Campania “Luigi Vanvitelli”, from June 2017 to July 2019.

We consecutively recruited the participants from the outpatient unit of the Internal Medicine Department. Before enrolment we obtained the written informed consent of the patients, who had previously received a full description of the study. The Institutional Review Board of University of Campania “Luigi Vanvitelli” approved the investigation protocol (study protocol number: 662/2017), in accordance with the updated Declaration of Helsinki for experiments involving humans.

We completed an ad hoc clinical history form at baseline to record the following patients’ sociodemographic and clinical characteristics—age, gender, marital and employment status, educational level, source of infection, duration of liver disease, previous unsuccessful therapies with interferon (IFN) or with ribavirin, family and personal history of psychiatric disorders, prior or current psychiatric diagnosis, active psychiatric treatment, prior psychiatric hospitalization, and history of substance abuse.

All recruited CHC patients received DAA treatment for the first time in accordance with the European Association for the Study of the Liver (EASL) Clinical Practice Guidelines and the standard of care in Italy. The DAA treatments administered are reported in Table 1.

Experienced psychiatrists interviewed recruited patients. In case of psychiatric diagnosis, the process of formulating was accomplished by applying the criteria of the Diagnostic and Statistical Manual of Mental Disorders, Fourth Edition (DSM-IV), and confirmed by the Structured Clinical Interview for DSM-IV, Axis I (SCID-I) and Axis II (SCID-II) [18].

### 2.2. Assessments

A total of 62 patients gave consent for hepatic and psychiatric evaluation before starting therapy with DAA agents. Seven patients denied consent.

#### 2.2.1. Hepatological Assessment

We diagnosed CHC through clinical examination, ultrasound scan, and liver function tests, and HCV infection through the positivity of serum anti-HCV and HCV-RNA. All patients underwent a complete physical examination and calculation of the body mass index (BMI: weight/height in meters^2^), tests of liver function, glycemia, hemoglobin A1c, total cholesterol, HDL, LDL, triglycerides, HCV genotype, and HBV plus HIV co-infection evaluation.

We assessed liver fibrosis and steatosis at baseline by a noninvasive method, namely transient elastography (TE) (FibroScan^®^, EchoSens, Paris, France), according to the EASL Clinical Practice Guidelines.

Furthermore, we performed biochemical tests, anti-HCV, and HCV RNA with kits routinely used in the hospital laboratory. We determined the HCV genotype using HCV genotype 2.0 assay (LiPA) (Bayer, Health Care, Eragny, France). Sustained virologic response 12 (SVR12) occurred when serum HCV RNA was below the lower detection limit, at least 12 weeks after the end of treatment [19]. Hereafter, by “after treatment,” we mean 12 weeks after DAA treatment completion. We followed up patients applying EASL Clinical Practice Guidelines [20].

#### 2.2.2. Psychopathological Assessments

We administered the following assessment instruments to recruited patients—the Hamilton Depression Rating Scale (HAM-D), the Hamilton Rating Scale for Anxiety (HAM-A), and the Symptom Checklist-90-Revised (SCL-90-R).

The HAM-D contains 21 items to evaluate somatic, cognitive, and emotional depressive symptoms. The scores range from 0 to 52, with the following cut-off scores—0–7 (absence of depression), 8–17 (mild depression), 18–29 (major depression), and 30–52 (severe depression) [21].

The HAM-A consists of 14 items assessing both psychic (mental agitation and psychological distress) and somatic anxiety (physical complaints). Each item corresponds to a set of symptoms grouped according to their nature and is scored on a 5-point scale ranging from 0 (not present) to 4 (severe), with a total score ranging between 0 and 56. The following total scores indicate different conditions of anxiety—≤ 17, mild, 18–24, mild to moderate, and 25–30, moderate to severe [22].

We used the SCL-90-R checklist, a multidimensional self-report symptom inventory, for screening and detecting clinical symptoms or indicators of psychological distress [23]. It includes 90 items using a 5-point level scale (from 1 = no problem to 5 = very serious) to measure the symptoms experienced in the last seven days. The items are grouped into nine subscales—(i) somatization, (ii) obsessive-compulsive, (iii) interpersonal sensitivity, (iv) depression, (v) anxiety, (vi) anger/hostility, (vii) phobic anxiety, (viii) paranoid ideation, (ix) psychoticism. A higher score on the SCL-90-R indicates more considerable psychological distress.

#### 2.2.3. Coping Strategies and Quality of Life Evaluation: COPE INVENTORY and SF-36

The Coping Orientation to Problems Experienced (COPE) Inventory [24] is a 60-item measure comprising 15 subscales with 4 items each, developed to assess the different coping strategies, such as *problem-focused* and *emotion-focused,* used in response to stress. *Problem-focused* coping strategies include active coping (taking steps to eliminate the problem), planning (thinking about dealing with the problem), suppression of competing activities (focusing only on the problem), restraint (waiting for the right moment to act), and use of instrumental social support (seeking advice from others). Emotion-focused strategies aim to reduce negative thoughts and feelings associated with the stressor and to manage the emotional distress associated with (or cued by) the situation. They include positive reinterpretation (framing the stressor in positive terms), acceptance (learning to accept the problem), denial (refusing to believe the problem is real), turning to religion (using faith for support), and emotional social support (seeking sympathy from others). The COPE Inventory includes further strategies, like the focus on and venting of emotions (wanting to express feelings), behavioral disengagement (giving up trying to deal with the problem), mental disengagement (distracting self from thinking about the problem), substance use (using alcohol or drugs to reduce distress), and humor (making light of the problem).

The Short Form 36 (SF-36) Health Survey includes 36 self-reported items grouped into the following eight dimensions—(i) vitality, (ii) physical functioning, (iii) bodily pain, (iv) general health perceptions, (v) physical role functioning, (vi) emotional role functioning, (vii) social role functioning, and (viii) mental health [25,26].

### 2.3. Statistical Analysis

We analyzed sociodemographic and clinical characteristics of the recruited patients using frequency counts, plus mean and standard deviations, as appropriate. We adopted nonparametric tests, considering the small sample size and the non-parametric distribution of continuous variables.

At baseline, we evaluated gender differences using Mann–Whitney test for independent samples.

We divided all CHC patients into two subgroups before data analysis—with (Group A) or without (Group B) a current and/or past psychiatric history. We, thus, examined the differences between clinician-rated and self-reported psychiatric symptoms, coping strategies, and QoL, before and 12 weeks after DAA therapy completion. We examined the differences between the two groups by a Mann–Whitney test for independent samples, and those within a group before and after therapy by a Wilcoxon test for paired samples, as appropriate. We also used Statistical Package for Social Sciences (SPSS) version 17.0 for the statistical analysis and set the level of statistical significance at *p* ≤ 0.05.

## 3. Results

### 3.1. Sociodemographic and Clinical Characteristics of Patients

In a total of 69 patients consecutively recruited from the outpatient unit of the Internal Medicine Department, 62 agreed to participate in the study, six patients refused to be assessed, and one reported a lack of time.

Sociodemographic and clinical characteristics of enrolled CHC patients are illustrated in Table 1. In our cohort, 43 patients (69.3%) had chronic hepatitis, and 19 (30.6%) cirrhosis. Group A and Group B included patients who were smokers (30.6% in both groups), overweight (38.7% vs. 40.5%) and obese (19.3% vs. 6.8%); also, with a sedentary lifestyle (35.5% vs. 42.0%) and an irregular dietary pattern (38.7% vs. 27.9%). Moreover, the patients showed comorbidity with hypertension (9.7% vs. 29.0%), diabetes type 2 (4.8% vs. 16.1%), renal failure (3.2%, only in Group B), and thyroid dysfunctions (4.8%, only in Group B). In both groups, the prevalent HCV genotype was Type 1 without HBV or HIV co-infection.

In Group A, a total of 21 CHC patients (33.9%) had a current and/or past psychiatric history, with the most frequent psychiatric diagnosis being the anxiety-depressive disorder (20.8%); 11 patients (17.7%) had received one or two psychotropic drugs before the study period, such as anxiolytics (*n* = 9, 14.5%), antidepressants (*n* = 5, 8.1%), and antipsychotics (*n* = 1, 1.6%). During DAA treatment, no patient required a psychiatric emergency visit nor hospitalization.

### 3.2. DAA Treatment

A total of 17 patients (27.4%) (Group A, *n* = 8; Group B, *n* = 9) had before received IFN-based therapies with or without ribavirin. The entire cohort was naïve for treatment with DAA agents, and after assessment received the following medications—sofosbuvir-velpatasvir (*n* = 13, 21.0% in Group A; *n* = 14, 22.6%, in Group B), ombitasvir/paritaprevir/ritonavir + dasabuvir (3D) (*n* = 3, 4.8%, in both Group A and B), glecaprevir/pibrentasvir (*n* = 7, 11.3% in Group A; *n* = 8, 12.9% in Group B), elbasvir/grazoprevir (*n* = 7, 11.3% in both Group A and B) (Table 1). The treatment was initiated at the recommended doses after the baseline evaluation.

All patients completed the treatment schedule and achieved an SVR12 (98.4%), apart from one patient, who relapsed at the end of treatment.

On the whole, treatment was well-tolerated and caused no altered liver enzymes, bilirubin or prothrombin time, and no sign of hepatic decompensation or renal impairment.

### 3.3. Psychopathological Status of CHC Patients before DAA Treatment

At baseline (T0), patients in Group A showed higher total scores of HAM-D (*p* < 0.05) and HAM-A (*p* < 0.05) compared to Group B (Table 2). Similarly, the SCL-90-R total score (*p* < 0.05), as well as the subscales of somatization (*p* < 0.05), depression (*p* < 0.05), anxiety (*p* < 0.05), paranoid ideation (*p* < 0.01), and psychoticism (*p* < 0.01) were higher in Group A (Table 2). No significant differences emerged from coping strategies adopted by patients of both groups before initiating DAA treatment (Table 3), as well as in the quality of life SF-36 subscale scores (Table 4), except for mental health subscale score, which was lower in Group A (*p* < 0.012). A preliminary analysis of gender differences in patients of both groups showed that in basal conditions, CHC male patients showed higher SCL-90-R subscale scores on somatization, obsessive-compulsive symptoms, depression, sensitivity, and psychoticism. Also, male CHC patients used coping strategies like denial, turning to religion, and humor less frequently.

No remarkable difference emerged from the baseline HCV RNA levels between the two groups, as well as between patients with and without cirrhosis.

### 3.4. Effects of DAA Treatment on Psychiatric Outcomes

In CHC patients of Group A, the HAM-A and HAM-D scores decreased significantly (*p* < 0.05). No difference emerged from total and subscale scores of SCL-90-R, as well as in coping strategies in Group A CHC patients (Table 3), with a significant decrease of general health perceptions SF-36 score (*p* < 0.05) (Table 4).

On the other hand, in Group B, both HAM-D and HAM-A scores significantly decreased (*p* < 0.0001), as well as the depression SCL-90-R subscale (*p* < 0.05), whereas somatization SCL-90-R subscale score (*p* < 0.001) remarkably increased (Table 2). Problem-focused COPE Inventory score notably increased in the suppression of competing activities (*p* < 0.01) and decreased in instrumental social support (*p* < 0.01), emotion-focused acceptance (*p* < 0.001) and mental disengagement subscales (*p* < 0.05) (Table 3). SF-36 scores improved in vitality (*p* < 0.01), physical role functioning (*p* < 0.05), and social role functioning (*p* < 0.05) in patients of Group B (Table 4). Besides, emotional (*p* < 0.037) and social (*p* < 0.007) role functioning, along with mental health (*p* < 0.037) SF-36 subscale scores, were higher than those of Group A patients after treatment (T1) (Table 4).

## 4. Discussion

Our study aimed to clarify whether DAA therapy may exacerbate mood symptoms in patients with a current and/or past psychiatric history or foster psychiatric symptoms in DAA drug-naïve CHC patients. We analyzed mental health alterations in a cohort of CHC patients before and 12 weeks after DAA therapy. Also, we explored in which manner such patients coped with the disease and complications, and how their quality of life was affected.

Our cohort of CHC patients treated with DAA agents achieved an SVR12 of 98.4%, a percentage similar to that reported in other studies [27,28]. We found no remarkable differences in the baseline HCV RNA level between patients with and without a psychiatric history, as well as between patients with and without cirrhosis.

Mental health problems frequently occur in both treated and untreated CHC patients, and depression appears as one of the most severe complications [2,28,29]. In our cohort, most CHC patients had a diagnosis of mixed anxiety-depressive disorder and were all DAA-naïve. Only 27.4% had been previously treated with IFN-α alone or in combination with ribavirin.

Our findings are consistent with those of other studies on the prevalence of depression and anxiety in CHC patients. Yet, differences exist in the design, sample, and size of the study, clinical and time evaluation, severity and definition of psychopathological symptoms, plus the use of self-reported or observer-rated diagnostic interview scales [29,30,31,32,33,34]. According to a few studies [12,35,36,37,38], DAA therapy is safe and does not worsen the depressive or other psychiatric symptoms in CHC patients with pre-existing mental diseases. Other studies, differently, feature the possible onset of mood symptoms in patients with psychiatric history [15,16,17].

We observed that DAA treatment neither worsened mood symptoms nor fostered the onset of new psychopathological symptoms in DAA-naïve CHC patients, regardless of current and/or past psychiatric history.

Our findings are in contrast with those deriving from previous studies suggesting that antiviral treatments may trigger worsened or newly arisen psychiatric symptoms [13,31,33]. In line with Sakamari et al. (2019) [16], only Group B patients reported a higher SCL-90-R score in somatization, when compared to their baseline conditions (Table 2). Self-reported depressive symptoms, instead, improved significantly (*p* < 0.02).

Overall, our analysis illustrates that a few patients of both CHC groups may prove to be more sensitive to some psychiatric adverse events associated with DAA therapy. Accordingly, a psychotherapeutic intervention would be needed to achieve a more effective outcome.

We used the COPE Inventory before and after therapy to explore how CHC patients coped with the disease and related complications. In particular, we found no remarkable differences in patients with pre-existing mental disorders before and after DAA treatment. Instead, patients from Group B significantly increased the use of problem-focused coping strategies for suppression of competing activities (*p* < 0.01) and decreased instrumental social support (*p* < 0.01). Oppositely, such patients significantly decreased emotion-focused strategies, such as acceptance (*p* < 0.001) and mental disengagement (*p* < 0.05). Therefore, our cohort of CHC patients without a psychiatric history seems to have higher cognitive flexibility and to be more adaptable to the new situation. Contrarily, in patients with a current or a pre-existing mental disorder, the psychological distress present at the baseline remains unchanged after DAA therapy. So, such patients, necessitate a cognitive-behavioral or interpersonal psychotherapy. In support of this argument, a recent study on CHC patients on antiviral therapy indicates that psychiatric counseling contributed to decreasing both the severity of psychiatric manifestations and the use of antidepressants or benzodiazepines [39].

HCV infection affects patients’ quality of life, social activities, and physical conditions. Such disease also causes psychological stress, work and financial burden, and compromises emotional well-being due to the fear of contagion and prognosis [40].

The SF 36-scores show a poor quality of life, as emerged in both our groups before treatment.

As already observed in other studies, psychiatric complications in CHC patients, as well as treatment with antiviral agents, may worsen QoL, daily activities, social and relational life [4,27,31,41]. Group B CHC patients improved a few physical-health related SF-36 scores, such as vitality and physical role functioning, along with social role functioning of mental-health-related subscale scores 12 weeks after completing DAA treatment. Group A patients, contrarily, reported no changes in SF-36 subscale scores after DAA therapy, except for general health perceptions subscale scores, which markedly decreased. They also showed a worse QoL when compared to Group B patients after DAA treatment (Table 4). Overall, our results show that DAA agents do not worsen QoL in CHC patients, contrarily to what occurred in IFN-α and/or ribavirin treated CHC patients, who displayed a marked decrease from the baseline SF-36 scores [13,42].

We limited our analysis to 12 weeks after completing DAA therapy considering the fact that SVR-12 is used as a primary efficacy endpoint. Moreover, we focused our study solely on neuropsychiatric side effects, rather than on the most frequent treatment-associated side effects like irritability, insomnia, asthenia, fatigue, dyspnea, and diarrhea. Psychiatric complications, in particular, may appear even 4 weeks after initiating antiviral therapy, with a higher intensity of depressive symptoms after 8 weeks [37,43,44]. Furthermore, a few neuropsychiatric side effects may persist until completion of antiviral therapy. More studies with lengthier follow-ups would be therefore needed to test if side effects may persist beyond the 12-week post-DAA-treatment period in correlation with the virologic response (i.e., SVR-24).

Moreover, we did not analyze possible interactions between DAA agents and other drugs that our CHC patients used for various medical comorbidities (i.e., oral antidiabetics, antihypertensive, and psychoactive drugs). Another limitation to acknowledge is that we solely recruited outpatients, which did not enable us to extend the generalizability of our findings to inpatients as well. The low number of participants and the prevalent diagnosis of mixed anxiety-depressive disorders did not allow us to analyze further the differences among subgroups with other psychiatric diagnoses. Previous studies centered mainly on patients with major depressive disorders and substance abuse or dependence [12,35]. Furthermore, the advanced liver disease in some patients, the relevant comorbidities, and no patients with HIV co-infection might act as impeding factors for achieving the generalizability of our results. Other limitations of our study are the absence of a control group and the inclusion of a higher number of patients in Group B due to the subdivision of recruited CHC patients with or without a current and/or past psychiatric history. Such factors may be responsible for a floor effect and explain the lower subscale scores that patients of Group B achieved in baseline conditions.

## 5. Conclusions

The EASL Clinical Practice Guidelines have recently established that the goals of the therapy for HCV infection [20] should include improving quality of life, eradicating stigma and providing psychological therapies to improve patient care.

A large number of studies have illustrated that in such types of patients, psychopathological symptoms can compromise interpersonal relationships, energy levels, body composition, mood and cognitive function and so damage the quality of life. Our study emphasized that in CHC patients with or without a psychiatric history, DAA therapy is safe and devoid of both hepatological and psychiatric side effects. Overall, DAA agents did not exacerbate or foster psychopathological symptoms in CHC patients naïve to antiviral therapy. We also showed that self-perceived distress may persist after DAA treatment in some patients of both CHC groups, suggesting that psychological interventions are needed to achieve better outcomes. Therefore, cooperation between hepatologists and psychiatrists/psychologists is suggested to achieve a more satisfying quality of life in CHC patients.

## Figures and Tables

**Table 1 brainsci-10-00483-t001:** General characteristics of enrolled chronic hepatitis C (CHC) patients with (Group A) and without (Group B) a current or lifetime psychiatric history in basal conditions.

Variables	Group A (*n* = 21)	Group B (*n* = 41)	*p*
Males (*n*, %)	7 (11.3%)	23 (37.1%)	
Age (Mean ± SD) (Range)	65.7 ± 9.2 (51–78)	62.0 ± 12.2 (33–82)	
BMI (Mean ± SD)	26.8 ± 4.5	26.1 ± 3.3	
Liver Fibrosis Score (*n*, %):			
0	1 (1.6%)	0 (0.0%)	
1	9 (14.5%)	12 (19.3%)	
2	3 (4.8%)	9 (14.5%)	
3	3 (4.8%)	6 (9.7%)	
4	5 (8.1%)	14 (22.6%)	
Median HCV RNA UI/mL × 10^6^ (Range)	2.87 (0.009–9.0)	3.54 (0.058–14.0)	
Genotype: 1	13 (21.0%)	29 (46.8%)	
2	5 (8.1%)	9 (14.5%)	
3	2 (3.2%)	3 (4.8%)	
4	1 (1.6%)	0 (0.0%)	
Previous Hepatocellular Carcinoma (HCC) (*n*, %)	1 (1.6%)	2 (3.2%)	
Concomitant Medical Diagnoses (*n*, %):			
Hypertension	6 (9.7%)	18 (29.0%)	
Diabetes Mellitus Type 2	3 (4.8%)	10 (16.1%)	
Renal Failure	0 (0.0%)	2 (3.2%)	
Thyroid Dysfunction	0 (0.0%)	3 (4.8%)	
Selected DAA Treatment (*n*, %):			
Sofosbuvir/Velpatasvir	13 (21.0%)	14 (22.6%)	
Ombitasvir/Paritaprevir/Ritonavir + Dasabuvir (3D)	3 (4.8%)	3 (4.8%)	
Glecaprevir/Pibrentasvir	7 (11.3%)	8 (12.9%)	
Elbasvir/Grazoprevir	7 (11.3%)	7 (11.3%)	

**Table 2 brainsci-10-00483-t002:** Median scores and ranges (min–max) of psychopathological scales in CHC patients with (Group A) and without (Group B) a current or lifetime psychiatric history, before (T0) and 12 weeks after completing treatment with direct-acting antiviral (DAA) agents (T1).

Psychopathological Scales	Group A (*n* = 21)			Group B (*n* = 41)		
	T0	T1	*p*	T0	T1	*p*
Hamilton Depression Rating Scale (HAM-D)	13.5 (2–25) ^a^	6.0 (0–23)	0.05	7.0 (0–19)	4.0 (0–17)	0.0001
Hamilton Anxiety Rating Scale (HAM-A)	13.5 (4–29) ^a^	6.0 (0–22)	0.05	9.0 (0–16)	4.0 (0–23)	0.0001
SCL-90-R: Total Score	43,0 (1–130) ^c^	60.0 (1–125)		22.0 (1–107)	23.5 (2–100)	
SCL-90-R: Subscales						
*Somatization*	6.0 (0–18) ^a^	10.0 (1–26)		4.0 (0–16)	5.0 (0–27)	0.001
*Obsessive-Compulsive*	9.0 (0–19)	7.0 (0–15)		3.0 (0–21)	3.0 (0–14)	
*Interpersonal Sensitivity*	4.5 (0–16)	6.0 (0–12)		2.0 (0–15)	2.0 (0–16)	
*Depression*	11.0 (0–29) ^a^	14.0 (0–27)		4.0 (0–34)	3.0 (0–21)	0.05
*Anxiety*	5.0 (0–20) ^a^	6.0 (0–22)		3.0 (0–15)	3.0 (0–13)	
*Anger-Hostility*	2.0 (0–11)	3.0 (0–10)		1.0 (0–6)	1.0 (0–8)	
*Phobic Anxiety*	0.0 (0–18)	1.0 (0–8)		0.0 (0–18)	0.0 (0–8)	
*Paranoid Ideation*	3.0 (0–11) ^b^	3.0 (0–10)		1.0 (0–10)	2.0 (0–12)	
*Psychoticism*	3.5 (0–11) ^b^	3.0 (0–10)		1.0 (0–8)	1.0 (0–7)	

^a^*p* < 0.05 vs. T0 of Group B; ^b^
*p* < 0.01 vs. T0 of Group B.

**Table 3 brainsci-10-00483-t003:** Median scores and ranges (min–max) of Coping Orientation to Problems Experienced (COPE) Inventory subscales in CHC patients with (Group A) and without (Group B) a current or lifetime psychiatric history, before (T0) and 12 weeks after completing treatment with direct-acting antiviral (DAA) agents (T1).

Cope Inventory Subscales	Group A (*n* = 21)			Group B (*n* = 41)		
	T0	T1	*p*	T0	T1	*p*
*Problem-Focused*						
Active Coping	13.0 (10–16)	13.0 (11–14)		13.0 (5–16)	12.5 (5–15)	
Planning	12.0 (9–16)	12.0 (8–15)		13.0 (5–16)	13.0 (5–16)	
Suppression of Competing Activities	8.5 (4–14)	12.0 (6–14)		9.0 (4–15)	11.0 (4–15)	0.01
Restraint Coping	10.0 (5–15)	12.0 (7–14)		11.0 (5–15)	10.0 (5–15)	
Instrumental Social Support	11.0 (4–159	12.0 (6–14)		12.0 (4–16)	10.0 (4–15)	0.01
*Emotion-Focused*						
Positive Reinterpretation	13.0 (10–16)	13.0 (11–14)		13.0 (5–16)	12.5 (5–15)	
Acceptance	13.0 (10–16)	11.0 (10–16)		14.0 (9–16)	11.5 (8–16)	0.001
Denial	6.0 (4–14)	5.5 (4–14)		7.0 (4–15)	5.0 (4–15)	
Turning to Religion	15.0 (4–16)	12.5 (4–16)		12.0 (4–16)	12.0 (4–16)	
Emotional Social Support	11.0 (4–15)	12.0 (6–14)		10.0 (4–16)	10.0 (4–15)	
*Others*						
Focus on and Venting Emotions	13.0 (6–16)	12.0 (7–16)		11.0 (4–16)	10.0 (6–16)	
Behavioral Disengagement	7.0 (4–13)	8.0 (4–13)		8.0 (4–12)	6.0 (4–12)	
Mental Disengagement	9.0 (5–16)	8.0 (5–13)		8.0 (4–14)	7.0 (4–14)	0.05
Substance Use	4.0 (4–10)	4.0 (4–10)		4.0 (4–16)	4.0 (4–7)	
Humor	7.0 (4–15)	6.5 (4–13)		7.0 (4–16)	5.0 (4–14)	

**Table 4 brainsci-10-00483-t004:** Median scores and ranges of SF-36 subscales in CHC patients with (Group A) and without (Group B) a current or lifetime psychiatric history, before (T0) and 12 weeks after completing treatment with direct-acting antiviral (DAA) agents (T1).

SF-36 Subscales	Group A (*n* = 21)			Group B (*n* = 41)		
	T0	T1	*p*	T0	T1	*p*
Vitality	50.0 (0–65)	55.0 (25–70)		52.5 (0–85)	60.0 (30–90)	0.01
Physical Functioning	95.0 (10–100)	95.0 (15–100)		95.0 (30–100)	92.5 (30–100)	
Bodily Pain	80.0 (20–100)	77.5 (22.5–100)		85.0 (10–100)	100.0 (22.5–100)	
General Health Perceptions	70.0 (5–85)	50.0 (25–80)	0.05	67.5 (10–100)	75.0 (30–95)	
Physical Role Functioning	100 (0–100)	90.0 (5–100)		80.0 (10–100)	100.0 (0–100)	0.05
Emotional Role Functioning	66.7 (0–100)	100.0 (0–100) ^c^		100.0 (0–100)	100.0 (0–100)	
Social Role Functioning	75.0 (0–100)	50.0 (12.5–87.5) ^b^		87.5 (12.5–100)	100.0 (50–100)	0.05
Mental Health	60.0 (4–76) ^a^	52.0 (28–729) ^c^		70.0 (20–92)	72.0 (36–96)	

^a^*p* < 0.012 vs. T0 of Group B; ^b^
*p* < 0.007 vs. T1 of Group B; ^c^
*p* < 0.037 vs. T1 of Group B.

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
