# Peer review of "Effects of Direct-Acting Antiviral Agents on the Mental Health of Patients with Chronic Hepatitis C: A Prospective Observational Study"

_brainsci, 2020, doi:10.3390/brainsci10080483_

Round 1
Reviewer 1 Report
The authors describe a prospective study where the effects of DAA on mental health is studied in patients with or without previous psychiatric issues who have HCC.
Minor issues-
1) What do you mean by drug naive patients? If you are administering DAA therapy then no patient is drug naive.
"What remains unclear is whether DAA therapy may exacerbate mood symptoms in patients with 64 prior and/or current psychiatric history or foster psychopathological symptoms in drug-naïve CHC 65 patients."
2)Page 4 line 175. Please clearly mention that these are group A patients.
3)Page 4 line 181-182- Were these IFN treated patients exclusively in Group A? Which cohort are you referring to?
Major issues-
The authors have over-extended their findings. In many metrics analyzed there seems to be no significant difference between the baseline and post-therapy data. The few metrics in a subset that have a significant p-value should be handled with tempered optimism. DAA therapy doesn't exacerbate the mental health issues of Group A but it doesn't help them either. In Group B the effects are not spectacular. The authors should consult a statistician for more robust tools to test the statistical significance of the data.
(See table 3 data): The values seems very close for To and T1 groups and yet the data is significant. In each main group eg emotion focused, only one or two criteria meet significance which cannot represent a true response.
Reviewer 2 Report
In this clinical study, Fabrazzo and colleagues investigated whether DAA treatment for HCV can affect the mental health of patients. The study reports that the treatment did not appear to cause any obvious side effects in the patients and may actually benefit psychological health. The study is very interesting. I am aware that males are more likely to have poor outcomes of HCV infection, compared to females. I think it would be interesting for the authors to analyze the difference between males and females in both group A and group B.
Round 2
Reviewer 1 Report
No comments at this time.